# Precision and Digital Agriculture: Adoption of Technologies and Perception of Brazilian Farmers

**Édson Luis Bolfe [1,2,](https://orcid.org) , Lúcio André de Castro Jorge [3](https://orcid.org) , Ieda Del'Arco Sanches [4](https://orcid.org) , Ariovaldo Luchiari Júnior [1], Cinthia Cabral da Costa [3], Daniel de Castro Victoria [1], Ricardo Yassushi Inamasu [3], Célia Regina Grego [1], Victor Rodrigues Ferreira [5] and Andrea Restrepo Ramirez [5]**

[1] Embrapa Informática Agropecuária, Brazilian Agricultural Research Corporation, Campinas 13083-886, Brazil; ariovaldo.luchiari@embrapa.br (A.L.J.); daniel.victoria@embrapa.br (D.d.C.V.); celia.grego@embrapa.br (C.R.G.)

[2] Department of Geography, Graduate Program in Geography, University of Campinas (Unicamp), Campinas 13083-885, Brazil

[3] Embrapa Instrumentação, Brazilian Agricultural Research Corporation, São Carlos 13560-970, Brazil; lucio.jorge@embrapa.br (L.A.d.C.J.); cinthia.costa@embrapa.br (C.C.d.C.); ricardo.inamasu@embrapa.br (R.Y.I.)

[4] Divisão de Sensoriamento Remoto, National Institute for Space Research (INPE), São José dos Campos 12227-010, Brazil; ieda.sanches@inpe.br

[5] Unidade de Competitividade do Sebrae Nacional, Brazilian Micro and Small Business Support Service (Sebrae), Brasília 70770-900, Brazil; victor.ferreira@sebrae.com.br (V.R.F.); andrea.ramirez@sebrae.com.br (A.R.R.)

\* Correspondence: edson.bolfe@embrapa.br; Tel.: +55-19-3211-5700

**Abstract:** The rapid population growth has driven the demand for more food, fiber, energy, and water, which is associated to an increase in the need to use natural resources in a more sustainable way. The use of precision agriculture machinery and equipment since the 1990s has provided important productive gains and maximized the use of agricultural inputs. The growing connectivity in the rural environment, in addition to its greater integration with data from sensor systems, remote sensors, equipment, and smartphones have paved the way for new concepts from the so-called Agriculture 4.0 or Digital Agriculture. This article presents the results of a survey carried out with 504 Brazilian farmers about the digital technologies in use, as well as current and future applications, perceived benefits, and challenges. The questionnaire was prepared, organized, and made available to the public through the online platform LimeSurvey and was available from 17 April to 2 June 2020. The primary data obtained for each question previously defined were consolidated and analyzed statistically. The results indicate that 84% of the interviewed farmers use at least one digital technology in their production system that differs according to technological complexity level. The main perceived benefit refers to the perception of increased productivity and the main challenges are the acquisition costs of machines, equipment, software, and connectivity. It is also noteworthy that 95% of farmers would like to learn more about new technologies to strengthen the agricultural development in their properties.

**Keywords:** agriculture 4.0; smart farming; farmer's attitudes; Brazil

## 1. Introduction

Digital innovation in agriculture represents, according to the United Nations Food and Agriculture Organization [1], a great opportunity to eradicate poverty and hunger and mitigate the effects of

climate change. Through digitalization, all parts of the agri-food production chain will be modified, since connectivity and the processing of large amounts of information in an instant allows for more efficient work, greater economic return, greater environmental benefits, and better working conditions in the field. However, implementing these changes will require governments to increasingly strengthen rural infrastructure and promote the development of rural communities [1] and small rural businesses, so that they can adopt and implement innovative solutions.

In this context of innovation, Digital Agriculture (DA) is part of the so-called "fourth industrial revolution", and its conceptual bases address aspects associated with Agriculture 4.0 [2], which derives from the Industry 4.0, and refers to the use of cutting-edge technology in food production. More recently, the term "Smart Farming" has also been used from the perspective of a development that emphasizes the use of information and communication technologies in the digital farm management cycle, through the intensive use of new technologies such as the Internet of Things, cloud computing, artificial intelligence, and big data [3]. In general, the conceptual basis of "Smart Farming" or "Digital Agriculture" comes from scientific knowledge, techniques, and equipment from Precision Agriculture [4] started from 1990s decade.

The digital agriculture can be understood and encompasses communication, information, and spatial analysis technologies that allow rural producers to plan, monitor, and manage the operational and strategic activities of the production system. In addition to the technologies already consolidated, such as field sensors [5–7], orbital remote sensors [8,9] and also embedded in UAV-Unmanned Aerial Vehicle [10], global positioning systems [11], telemetry and automation [12], digital maps-soil relief, production, productivity [12], digital agriculture also involves the Internet and connectivity in crops [13,14], cloud computing, big data, blockchain and cryptography [3,15,16], deep learning [17–19], Internet of Things (IoT) [20], mobile applications and digital platforms [21,22], and artificial intelligence [23]. All these technologies support pre and post-production decisions and greater sustainability of production systems [24,25], in addition to access to a differentiated market benefiting short marketing chains.

The perspectives of American rural producers were assessed for benefits of using precision agriculture technologies, observing that the perceptions of the derived benefits are heterogeneous and differentiated according to the agricultural culture [26]. The authors point out that in order to better understand farmers' adoption decisions, or lack of, it is important to first understand their perceptions of the benefits that technologies provide. Reference [27] emphasize that digital agriculture supports better decision-making based on consistent analyzes of agricultural systems, supporting the farmer in the form of digital solutions associated with robotics and artificial intelligence. However, they stress that it is necessary to coordinate more solid user training, especially for young farmers eager to learn and apply modern agricultural technologies and to grant a generational renewal yet to come. They consider the right time for society to advance in a modern and sustainable agriculture, capable of presenting all the power of agricultural management based on data to face the challenges posed to food production in the 21st century.

Projections from Brazil's Ministry of Agriculture, Livestock, and Supply [28] indicate that grain production may rise from the current 250.9 million tons in 2019/2020 (65.6 million hectares) to 318.3 million tons in the 2029/2030 harvest (76.4 million hectares), corresponding to an increase of 26.9%; that is, a 2.4% growth per year. For meat production (beef, pork, and poultry), projections indicate that it should rise from 28.2 million tons to 34.9 million tons by the end of the next decade, representing a 23.8% increase in the period. Precision agriculture and the digital transformation that has occurred in rural areas can contribute to Brazil reaching or exceeding these expectations, strengthening the country's position as one of the World leaders in food production and export, based on increased productivity and a sustainable use of natural resources.

Reference [29] investigated the adoption of precision agriculture technologies in sugarcane production in the State of São Paulo, through a questionnaire sent to all companies that operate in the sugar and alcohol sector in the region. The authors concluded that companies that adopted and

used these technologies have proven to reap benefits, such as management improvement, higher productivity, lower costs, minimization of environmental impacts, and improvements sugarcane quality. Reference [30] investigated the use and adoption by producers and service providers on precision agriculture technologies in different Brazilian agricultural regions and found that the growth in technology adopted was linked to economic gains in agriculture. Economic aspects combined with the difficulty in using software and equipment provided by the lack of technical training by field teams were highlighted as the main factors that limit the expansion in the use of these technologies in the field.

Precision agriculture is already a reality for Brazilian technicians and rural producers. The knowledge that there is a variability in the production areas according to soil variations, vegetation, and the history of land use, is spreading progressively [31]. However, there are still important gaps in studies on a national scale to support strategic decisions in the development of new research, innovation, and the market. It is necessary to conduct studies in the context of digital agriculture in Brazil on aspects such as which technologies and applications are used, the perception of benefits, and the main challenges and expectations.

The future of using decision support systems in Agriculture 4.0 lies in the researchers' ability to better understand the challenges of this decision-making, including its applications in planning agricultural activities, managing water resources, adapting to climate change and food waste control [32]. The literature review shows different thematic clusters of extant social science on digitalization in agriculture: (i) adoption, uses, and adaptation of digital technologies on farm; (ii) effects of digitalization on farmer identity, farmer skills, and farm work; (iii) power, ownership, privacy and ethics in digitalizing agricultural production systems and value chains; (iv) digitalization and agricultural knowledge and innovation systems; and (v) economics and management of digitalized agricultural production systems and value chains [33]. This future research agenda provides ample scope for future interdisciplinary and transdisciplinary science on precision farming, digital agriculture, smart farming, and agriculture 4.0. Thus, this work aimed to gather information through an online consultation with rural producers about the digital technologies used in Brazil today, their applications, challenges, and future perspectives.

## 2. Materials and Methods

For the researcher's methodological definition, it was based on aspects applied: (i) in the evaluation of the factors of adoption of remote sensing images in the precision management of cotton producers in the United States [34]; (ii) in assessing the adoption and future prospects of precision agriculture in Germany [35]; (iii) in who evaluated the adoption of precision agriculture technologies in the sugarcane sector in Brazil [29], (iv) in who evaluated the adoption of precision agriculture under the perception of farmers and service providers in Brazil [30], and (v) in who evaluated different perspectives of American rural producers regarding the benefits of precision agriculture [26]. From the bibliographic references and the experience of the project team in the Brazilian context, specific questions were established, addressing aspects with the possibility of multiple choice answers on the technologies in precision agriculture and digital agriculture used in different complexity of applications (Table 1), among them in different sectors of agriculture and profiles of rural producers, added to perceptions of benefits, challenges and future expectations in Brazilian agriculture.

**Table 1.** Technologies in precision agriculture and digital agriculture in the context of the present study.

| Technologies | Reference | Complexity of Applications |
|---|---|---|
| Internet and Connectivity/Wireless | [14,36] | |
| Mobile APPs, Digital Platforms and Software | [21,22] | Low |
| Global Positioning Systems | [11] | |
| Digital Maps | [12] | |

**Table 1.** *Cont*.

| Technologies | Reference | Complexity of Applications |
|---|---|---|
| Proximal and Field Sensors | [5–7] | Medium |
| Remote Sensors | [8–10] | |
| Embedded Electronics, Telemetry, and Automation | [12,37] | |
| Deep Learning and Internet of Things | [17–20] | High |
| Cloud Computing, Big Data, Blockchain and Cryptography | [3,15,16] | |
| Artificial Intelligence | [23] | |

The questionnaire was prepared, organized, and made available to the public through the online platform LimeSurvey under the registration number 25889/2020 [38]. The system was available from 17 April to 2 June 2020, allowing answers to be collected based on specific questions previously defined (Table 2). The based used for the sample the research, was the Mailing List of Embrapa (Brazilian Agricultural Research Corporation, Campinas, Brazil) and Sebrae (Brazilian Micro and Small Business Support Service, Brasília, Brazil), agricultural cooperatives and associations of rural producers. A similar approach to sample respondents was applied in [30]. An "invitation message" was sent explaining the objectives with an access link and other guidelines to those interested in voluntarily participating in this online survey.

**Table 2.** Questions asked in precision agriculture and digital agriculture in the context of the research.

| Questions |
|---|
| What is the identification and location of the responding farmer? |
| In which productive sector does he/she operate in? |
| For how many years have you been working in this job? |
| What is the cultivated area? |
| What techniques and inputs are used in the agricultural production? |
| What technologies in digital agriculture were used? |
| What are the main functions of the technologies used in digital agriculture? |
| How do you access the technology used by digital agriculture in on your property? |
| Is it by cell phone apps, machines, equipment, data or images? |
| What are your perceptions on the advantages enabled by using technologies from digital agriculture? |
| What are the difficulties in accessing and using technologies from digital agriculture? |
| In what applications would you like to start or strengthen the use of digital technologies in the future? |

The primary data obtained for each question (single choice, multiple choice, and matrix questions) and its respective complete answers were consolidated in a LimeSurvey platform report and later exported in csv and included in a spreadsheet. Subsequently, statistics were generated based on absolute frequency data and graphs representative of the relative frequency represented by the percentages of each of the variables associated with the survey questions. Considering the size of the sample population of rural properties in Brazil approximately 1.5 million farmers in the survey profile, [39], it was stipulated to reach at least 385 questionnaires answered in full to obtain up to 5% margin of error with a 95% reliability level. However, results were gathered for 504 farmers (0.03% of population), a higher response rate than originally expected.

## 3. Results and Discussion

The consultation was nationwide and a total of 504 questionnaires were answered in full, including 154 (30.6%) from the South; 150 (29.8%) were from the Southeast region; 137 (27.2%) from the Northeast; 39 (7.7%) from the Midwest; and 24 (4.8%) in the North (Figure 1). The five states with the highest number of respondents were: Rio Grande do Sul (18.9%), Minas Gerais (13.9%), São Paulo (11.9%),

Bahia (11.1%), and Paraná (7.9%), which represent 61.7% of the respondents. These States are part of consolidated agricultural regions and, together with Mato Grosso, Goiás, and Mato Grosso do Sul, are part of the eight States with the highest gross agricultural production value in Brazil [40].

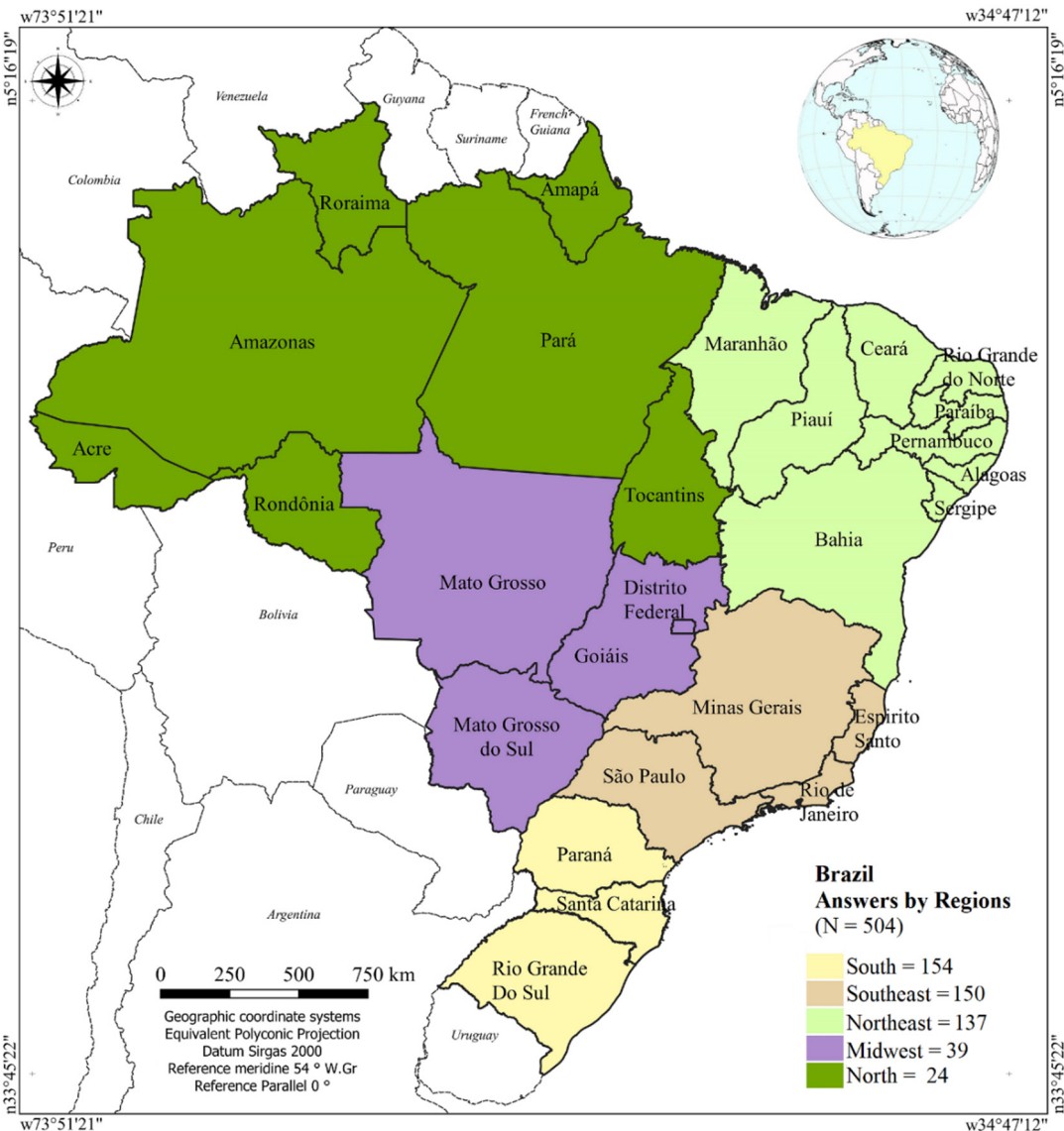

**Figure 1.** Distribution in percentage of farmers participating in the survey by Brazilian region.

Among the rural producers who participated in the survey, 74% work in agriculture (grains, fruit, horticulture, etc.); 54% with livestock (beef, pork, poultry, etc.); 6% with forestry (eucalyptus, pine, native, etc.) and 14% with other activities such as beekeeping, floriculture, aquaculture, and fish farming—not exclusively (Figure 2a); 72% cultivate areas of up to 50 hectares (Figure 2b); and 69% declared that they have more than 10 years of experience (Figure 2c). The percentage of respondents with areas of up to 50 hectares is in line with the distribution in size of properties in Brazil. Data from the last Brazilian Agricultural Census shows that 81.5% of rural establishments have less than 50 hectares, 15% between 50 and 500 hectares, and 2% above between 500 and 10,000 hectares of area, of which 46.7% work exclusively with agriculture, 48.8% with livestock, and 3.8% with forestry [39].

Half of the farmers participating in the survey use chemical inputs and controls; 43% crop rotation or pasture; 37% organic inputs and biological controls; and 24% intercropped or integrated systems, such as crop–livestock–forest integration systems or agroforestry systems (Figure 2d). The cultivation

of 53 types of perennial and temporary crops was reported by farmers, the main ones being corn, beans, soybeans, coffee, sugar cane, cassava, wheat, rice, vegetables, and fruits, especially banana, orange, grape, papaya, and mango. These crops represent the main Brazilian commodities, according to crop data for the years 2019/2020 [28].

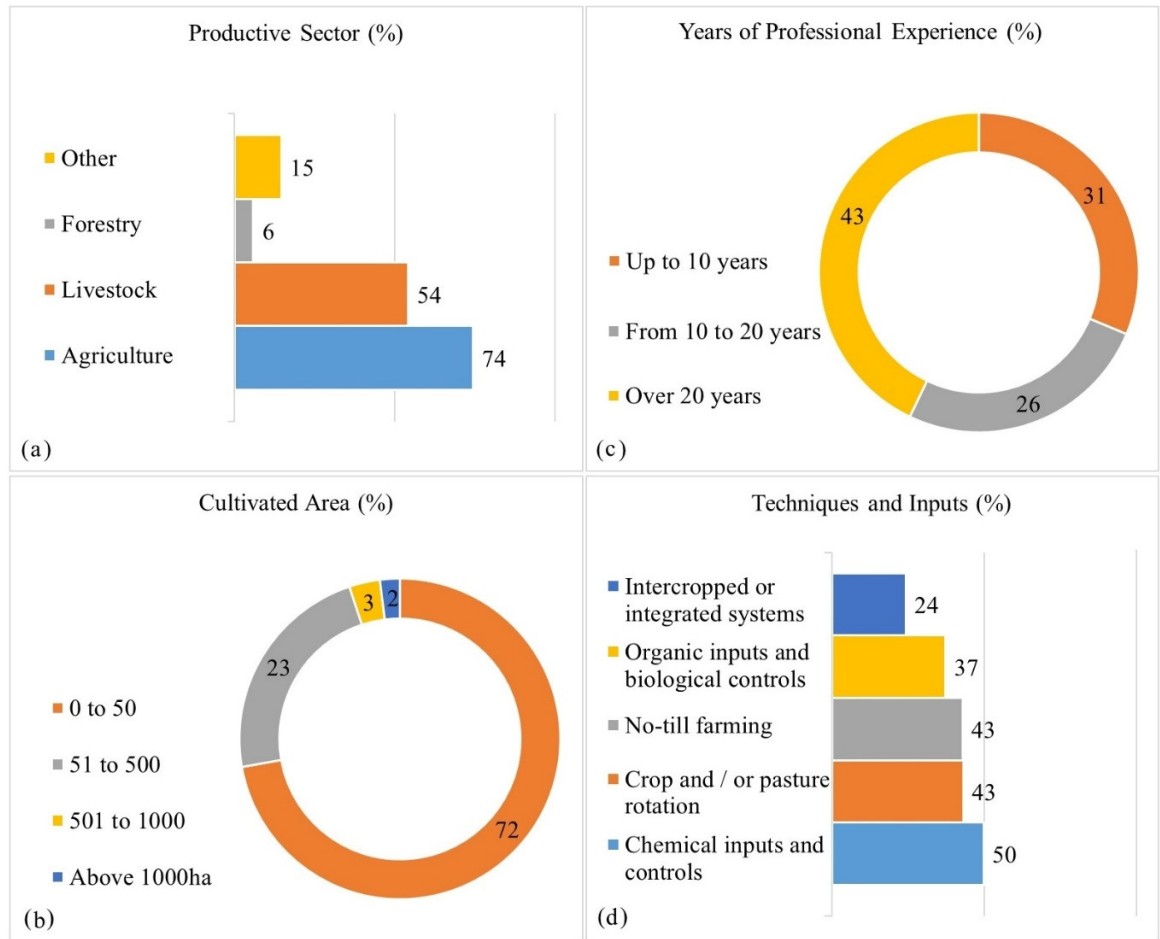

**Figure 2.** Profile of the Brazilian farmers that participated in the survey. Productive Sector (**a**); Cultivated Area (**b**); Years of Professional Experience (**c**); and Techniques and Inputs (**d**).

When interpreting the results presented, it is important to keep in mind that they are associated with a research by sample, which means it represents the different profiles of Brazilian farmers participating in the research. This is particularly important, considering the regional differences in relation to the socioeconomic and structural profile, where properties with larger dimensions and with a higher level of technification tend to show a higher level in the use of digital technologies. Therefore, caution must be taken when extrapolating or generalizing the results presented here to all Brazilian profiles and regions.

In reference to the digital technologies used, 15.9% of rural producers indicated that they still do not use any of these technologies; that is, 84.1% use at least one of the technologies listed in their production process (Figure 3). Among the technologies used, mainly low complexity ones, involving Internet access and connectivity/wireless on the property (70.4%), mobile applications, digital platforms, and software to obtain general information (57.5%), stand out. Reference [21], highlight that there is a growing increase in smartphone applications available to improve farmers' decision-making, where 95% of them already use a smartphone and out of these, 71% have specific applications that provide information about a specific culture, and detection and prediction of pests or diseases.

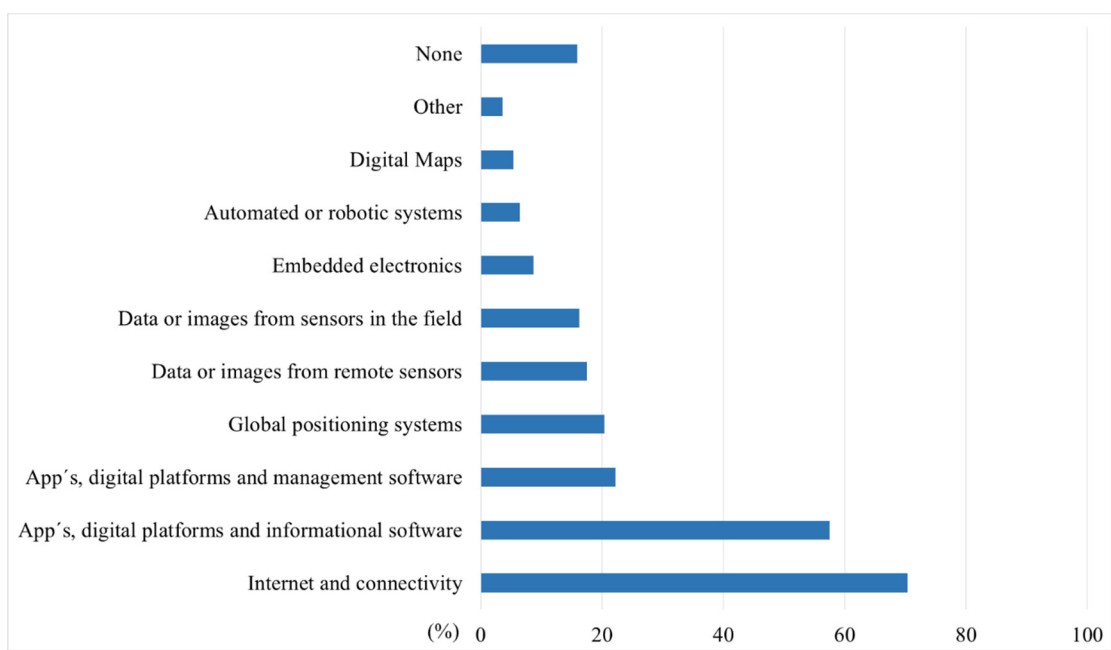

**Figure 3.** Precision and digital agriculture technologies used by farmers.

Medium complexity technologies, mobile applications, digital platforms, and management software (22.2%), global positioning systems (20.4%), data or images from remote sensors-satellite, airplane, UAV (17.5%), and field sensors-plant, animal, soil, water, climate, disease, or pests (16.3%) are used by a smaller portion of farmers. Here, the consolidated Global Positioning System (GPS) technology stands out, which since the beginning of the implementation of precision agriculture, has supported an increasing range of activities and generated countless benefits in rural areas. The applications of GPS in supporting soil sampling and data collection, enabling the analysis of spatial variation, improving the navigation process of machines and equipment for planting and handling crops, minimizing the use of agricultural inputs, in the possibility of working in low visibility field conditions, such as rain, dust, fog and darkness, and in supporting the generation of agricultural productivity maps and increasing the efficiency of aerial spraying [11].

The more complex technologies associated with machines or equipment with embedded electronics, e.g., autopilot, telemetry, and applications at varying rates (8.7%), automated or robotic systems (6.5%) and digital maps, e.g., spatial variability of soil, relief, harvest monitor, vegetation indexes, and productivity (5.4%), with high potential for economic return, present a relatively low percentage use by the interviewed farmers in relation to the other technologies. The greater the soil spatial variation in the property and its influence on the application of inputs, the greater the potential economic return of the application at a variable rate in comparison with the uniform applications of the traditional system [12]. That is, the variable application of inputs depends on many factors; however, the authors commit that the variability inherent in an agricultural soil and the relative responsiveness of the yield to fertilizer inputs at different levels of concentration in that soil are the most important factors to influence the economic gain.

When evaluating the use of precision agriculture technologies, specifically in the sugar cane sector in São Paulo, the main technologies pointed out were the use of satellite images (76%), autopilot systems (39%), and applications of inputs at a variable rate (29%) [29]. In the Brazilian states of Rio Grande do Sul, Paraná, Maranhão, Goiás, and Tocantins, 67% of soybean, corn, coffee, and cotton farmers used some technology in precision agriculture in the 2011/2012 harvest, with 56% using GPS mapping and 22% using remote sensors (satellite or airplane images), field sensors, and telemetry [30].

When analyzing functions of digital technologies used by farmers nowadays, a first group (between 40% and 65%) presented a broad and non-specific use of applications within production;

that is, applications focused on obtaining general information, property management, acquisition of inputs, and in trading a production or specific products (Figure 4). A second group of applications (between 20% and 40%) is focused on mapping and planning the use of land, predicting climatic risks, such as frost, hail, summer, and intense rain, depending on the Brazilian region, implementing animal welfare processes, and obtaining estimates of agricultural production and productivity. A third group of applications (below 20%), mostly related to processes with a higher technological complexity level, involves processes of detecting the control of nutritional deficit, diseases, pests, weeds, operational failures, and water deficit, in addition to applications associated with certifications and traceability of agricultural products.

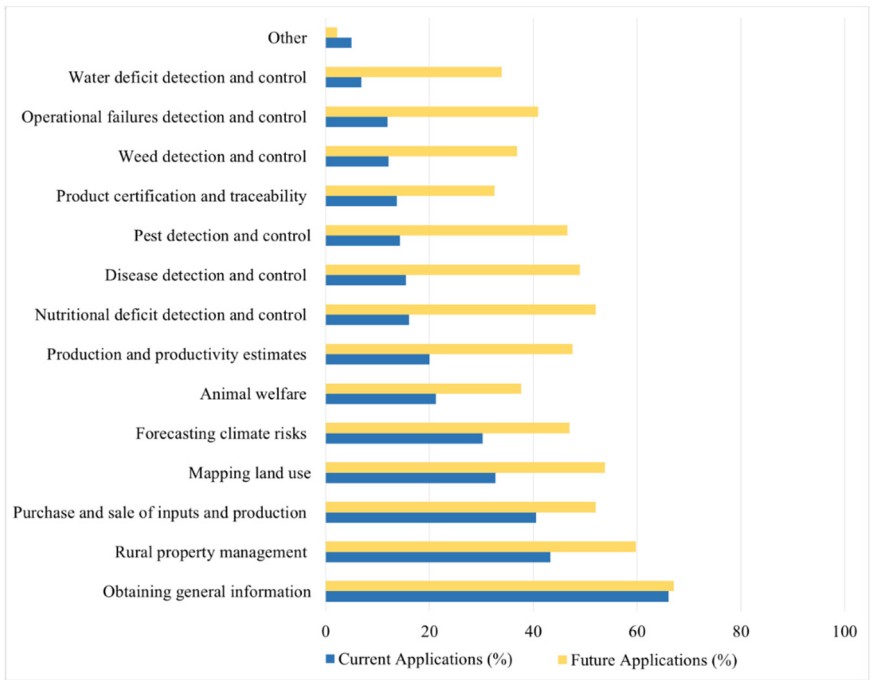

**Figure 4.** Farmer's current and future applications in precision and digital agriculture.

When analyzing future expectations, it was observed that farmers have an interest in initiating or strengthening the use of all the applications previously described (Figure 4). However, the most important topics in the percentage difference between current applications and future applications are in more complex activities involving the detection in the control of nutritional deficiencies (35.9%), diseases (33.5%), pests (32.2%), operational failures (29.0%), water deficit (27%), and weeds (24.8%). The applications for production and productivity estimates (27.6%) and in the mapping of land use (21.1%) are also noteworthy.

The prospect of increasing the applications of greater complexity observed is related to the growing evolution of remote sensors-orbital or field and data processing algorithms such as artificial neural networks. In remote sensing, the possibilities are highly diversified by increasing the spatial resolution, reaching the sub-metrical level, increasing the temporal resolution, with daily revisits, and better spectral resolution, with dozens of bands at different wavelengths of electromagnetic radiation. The sensors, initially focused on the visible or near infrared bands, today can range from ultraviolet to microwave, allowing even more advanced applications via hyperspectral sensors, light detection and ranging (LiDAR), fluorescence spectroscopy, and thermal spectroscopy. These conditions have driven the applications of greater complexity, allowing the evaluation of the properties of the soil and of an agricultural culture through analysis of specific compounds, molecular interactions and biophysical or biochemical characteristics [8]. There are also growing applications of spectral indexes of vegetation, such as the traditional NDVI (Normalized Difference Vegetation Index) and the EVI

(Enhanced Vegetation Index), adjusted version of NDVI with a focus on more dense vegetation, which allow management decision making of soil or agricultural crops in almost in real time.

In research with artificial neural networks to predict the harvest area, productivity, and production for soybean in Brazil by gathering data from a 1961–2016 time series, Reference [41] obtained reliable models for future predictions and support to farmers and the market in anticipating productive information. In the analysis carried [30], a large percentage of farmers in Brazil pointed to the expectation of implementing more complex and advanced technologies, such as sensor systems, to support the planting and application of fertilizers at different rates (67%), analysis, and integration of different databases in agriculture (78%).

The approach to access the technologies used by farmers was also addressed in a specific question, with farmers indicating that it occurs through direct acquisition and own use of machines, equipment and applications (68.8%); access through consultancy or services offered by associations, cooperatives, unions, NGOs (30.8%); access through consultancy or public services offered by city halls, state or federal government (20.8%); and via contracting services or consultancies specialized in digital agriculture (19.0%). Thinking about the future, it should be noted that 95% of the total respondents indicated that they were interested in receiving more information about digital agriculture and its applications.

Another aspect raised in the research was the perception of farmers about the positive impacts obtained in their production process considering the use of digital technologies aggregated in three groups: (i). images from remote sensing; (ii). field sensors, machinery, and equipment, and (iii). mobile applications, digital platforms, and software (Figure 5). All the impacts indicated were above 50% positive for farmers. This wide range of positive perceptions may be associated with the breadth of possibilities for the application of digital technologies throughout the farmer's production process. As highlights, there was a perception of increased agricultural productivity, pointed out by 64.7% of farmers, followed by perceptions of easier marketing and better planning of the daily activities of the property (62.7%); the reduction in production costs (62.3%) and the increase in the profit obtained (60.9%). The result indicates that the potential increase in agricultural productivity associated with commercialization is a key factor in managerial decision-making on the applications of digital technologies.

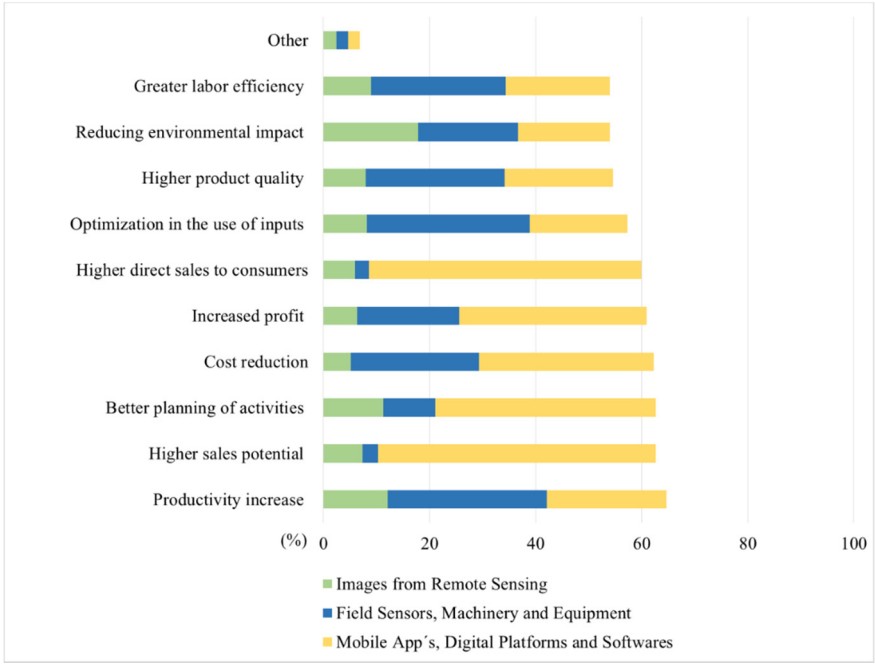

**Figure 5.** Farmer's perceptions about the advantages provided using precision and digital agriculture.

When assessing the perceived benefits of using four technologies in precision agriculture (variable rate fertilizer application, precision soil sampling, guidance and autosteer, and yield monitoring) with American farmers, also noted that the main perceived benefit is the increased productivity, followed by reduced production costs and greater convenience [26].

When analyzing individually the technologies adopted, in face of farmer's perception of benefits in the production system, it indicated that the use of mobile applications, digital platforms, and software is the group of technologies with the highest perception in gains, especially for increasing trade, greater direct sales with the consumer and to better plan the daily activities in the property. These results suggest that farmers are perceiving the marketing channels as an important aspect to leverage and implement a greater profit in the post-production process. An approach that is amplified by the greater smartphone dissemination in rural areas and "short marketing circuits" channels in the production based on the direct interaction between rural producer and urban consumer. In this sense, digital technologies in the management area that guide the integration of technical data with financial and management can collaborate in the decision-making of producers and in the planning of the rural business.

Field sensors, machinery, and equipment, on the other hand, was the group of technologies associated with a greater perception of positive impact on the optimization of the use of agricultural inputs, increased productivity and higher quality of products generated on the rural property. The results suggest that farmers seek to reduce production costs and/or increase yields for a more profitable system based on a better management of the agricultural input application process, probably associated with prior information from field sensors and variable rate applications, made possible by machines and equipment connected to global satellite positioning systems. Already emphasized that yield and production quality must be maximized, while the use of inputs, such as seeds, fertilizers, herbicides, and fungicides, must be optimized or minimized, pointing out the importance of using field sensors in these applications [5]. Thus, it is possible to maximize the economic return of agricultural production units through the measurement of traditional information, such as grain yield and moisture content, and new applications, such as grain protein content and straw yield.

The group that uses images from remote sensing had the lowest level of perception of positive impact in relation to the other groups, with a certain emphasis on the reduction of environmental impact. The greatest indication in the environmental issue, may be associated with the well-known and well-established use of images for mapping and monitoring land usage and coverage in Brazil. These issues are linked to the Brazilian Forest Code, which from 2012 made it mandatory to prove the maintenance of forest areas within the properties, ranging from 20% to 80%, depending on the region.

The developing new remote sensors in satellites, nanosatellites, and UAVs for agriculture have significantly enhance the potential of the applications in last year's [8–10], e.g., soil assessment, field mapping and monitoring-variable rate application, fertility, irrigation and drainage, harvest planning, and livestock monitoring. Further, new applications are developed with customized remote sensors with machine learning algorithms and computer vision to gather more accurate data and imagery. However, even with the increasing availability of images, the agricultural sector is yet to implement remote sensing technologies fully due to knowledge gaps on their sufficiency, appropriateness, and techno-economic feasibilities [42]. It may also be associated with the smaller number of farmers using remote sensing in relation to the other two groups of technologies evaluated.

Analyzing precision technologies with a focus on sustainability, Reference [43] highlights that, with its applications, the ability to identify spatial variability within the field increases and the use of this information for a more assertive crop management, operating resources with greater efficiency, making agriculture more productive, more sustainable, and more environmentally friendly. Attitudes like of confidence toward using the technologies, perceptions of net benefit, farm size, and farmer educational levels positively influenced the intention to adopt precision agriculture technologies [44].

In attempt to better understand the challenges and deterrents to the growth and expansion of digital technologies used by farmers, the survey also addressed to respondents what would be the

main difficulties faced. These challenges were evaluated at the aggregate level of technologies in rural properties; that is, specific technologies were not assessed (Figure 6). Many of these challenges are related to the cost of digital technologies, be it purchase, upgrade or acquiring services. It was observed that even with the relative high use of digital technologies, indicated by the study (Figures 3 and 4), there are still important challenges or barriers, mainly associated with the investment value for the acquisition of machines, equipment or applications (67.1%); problems or lack of connection in rural areas (47.8%), the value for hiring specialized service providers (44%); and the lack of knowledge about which technologies are most appropriate to be used in their property (40.9%) (Figure 6). A similar analysis carried out in the United States indicated that the main barriers pointed out by farmers are also related to the costs associated with precision agriculture technologies and services [45].

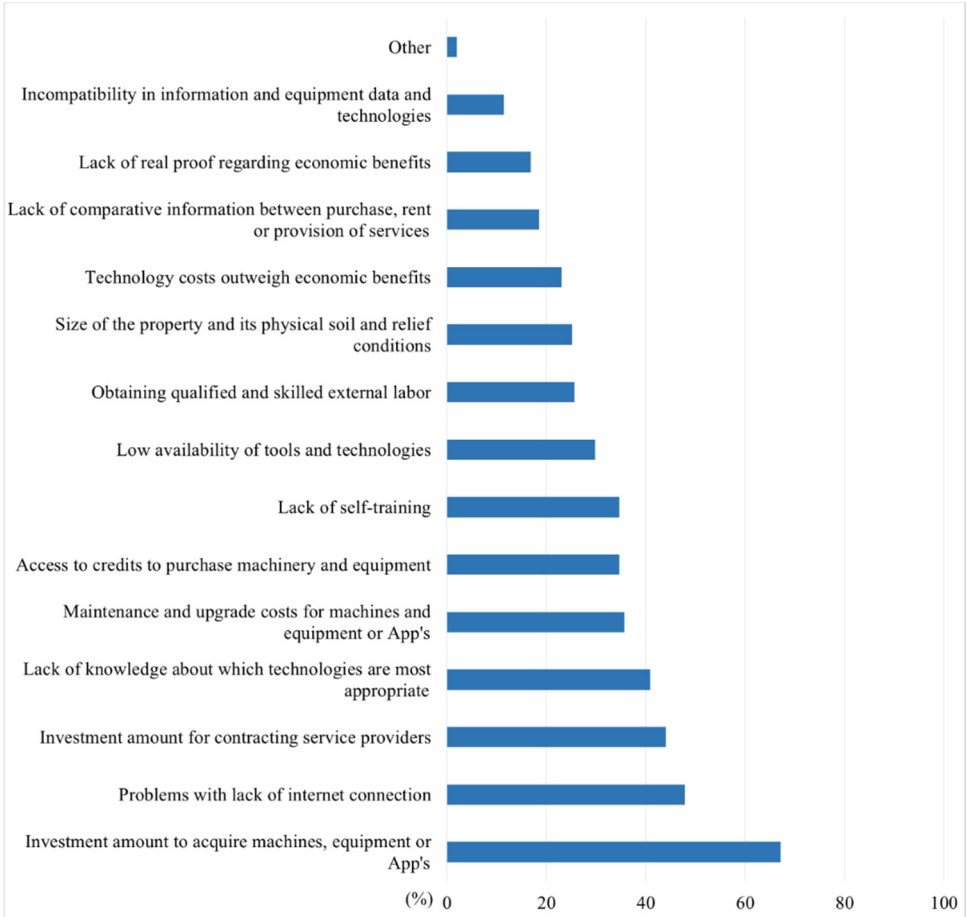

**Figure 6.** Difficulties and challenges farmers face to strengthen the use of precision and digital agriculture.

When analyzing the questions related to farmers investment considering only smaller farms (>20 ha), it was observed that 68.5% indicated the value for the acquisition of machines, equipment, or applications; and 45.3% indicated the value for hiring specialized service providers as the main difficulties and challenges. Thus, acquisition costs are considered a large difficulty/challenge across all farm sizes. However, it is interesting to note that for properties above 500 ha, despite large percentage indicating acquisition costs as a challenge (66%), connection issues were the main answer (69%). This difference is even more pronounced on larger properties (>1000 ha), where connection issues were indicated by 69% of the respondents and acquisition costs by 54%, still a large percentage.

The high percentage highlights difficulties in the acquisition, service value, and access to credits can be related, mainly, to the fact that currently the technologies available are mostly imported,

thus indicating a significant space for investment in the national industry in digital technologies, making the acquisition process more accessible. Brazil has developed important regions in digital technologies; however, the scarcity of economic resources in many properties and the small number of public research organizations in the sector can inhibit decisions to adopt these technologies [37].

It should be noted that there are already several applications and free software on the market that can partially meet this demand. Strategic actions to make these public APPs solutions available to different production systems-animal or plant, can raise the interest and use of digital technologies even further [45,46]. In addition, the strategy of courses, training and availability of content that strengthens knowledge management can also contribute to the greater use of specific digital technologies.

Even though Brazil is among the ten main world markets for mobile telecommunications and fixed broadband, according to an analysis by the National Telecommunication Agency of Brazil [47], the last Agricultural Census pointed out that access to the Internet covers only about 30% of farmers [39]. The territorial dimensions, the low demographic density of a large part of the rural environment and the socioeconomic inequalities are some of the main obstacles to increase the access to Internet in the country. The National Bank for Economic and Social Development of Brazil [48] estimates that greater connectivity in agriculture through the Internet of Things (IoTs) could generate 50 to 200 billion dollars of annual economic impact in 2025. Thus, it is strategic that public and private actions are implemented to expand the availability of the Internet and connectivity in the field.

It is also noteworthy that only 16.9% of farmers pointed out that the lack of proof of economic benefits is a challenge to strengthen the use of digital technologies in Brazil. Result that reflects the high perception in potential impacts and future benefits of digital technologies that farmers have underscored in this research. In this sense, "The future of Smart Farming may unravel in a continuum of two extreme scenarios: (1) closed, proprietary systems in which the farmer is part of a highly integrated food supply chain or (2) open, collaborative systems in which the farmer and every other stakeholder in the chain network is flexible in choosing business partners as well for the technology as for the food production side. The further development of data and application infrastructures (platforms and standards) and their institutional embedment will play a crucial role in the battle between these scenarios" [3] (p. 69).

The main drivers of adopting digital agriculture technologies in Brazil were increased productivity, better process quality, reduced costs, and greater knowledge of cultivated areas [49]. The use of technologies such as GPS, field sensors, remote sensors and telemetry could grow rapidly in areas with higher value-added crops, such as citrus and sugar cane, in States with higher land value, and regions with a strong base of Brazilian agricultural research organizations [37]. Some indicators suggest that the availability of sensors, mapping technology, and tracking technologies have changed many farming systems and hat these technologies will lead to relevant analysis at every stage of the agricultural value chain-from producers to consumers [50].

The rapid development of the Internet services, and, consequently, the greater number of instruments and devices connected to the network, increased the amount of data produced and collected every day in rural areas. The land structure with many small, medium, and large-scale Brazilian properties, and the relief conditions, especially the Cerrado Biome and the news forms of market access directly with the consumer should further strengthen the use of digital technologies in the medium term and support Brazilian rural development.

In addition to the technologies already consolidated in precision agriculture in Brazil, such as field sensors, remote sensors, embedded electronics, telemetry and automation, digital agriculture also involves the use of mobile apps, digital platforms, social networks, big data, Internet of Things, artificial intelligence, cloud computing, blockchain and cryptography, allowing to support greater sustainability of production systems. The use of digital technologies, underscored by Brazilian farmers, are differentiated and currently cover a wide spectrum, mainly in obtaining information about property management, product sales, inputs acquisition, property mapping and climate forecasting, presenting

future interests in a greater use in production estimation, detection and control in factors related to soil physic-biotics factors, and other stresses of the agricultural crops.

It is important that the public and private sectors evaluate their forms of aid to enhance the use of these digital technologies, since a large proportion of farmers, especially those with lower technical skills, still need support and technical monitoring to adapt and become more familiar with the use and applications of these technologies. Moreover, it is essential to structure a support process to enable the producer's decision making by presenting the costs and benefits in the acquisition and use of such technologies.

## 4. Conclusions

The digital agriculture can be understood as a set of technologies for communication, information, and analysis that allows farmers to plan, monitor, and manage the operational and strategic activities of the agriculture production systems, from pre-production, in production, and post-production. The results observed in the survey indicate that 84% of the Brazilian farmers interviewed use at least one digital technology in their production system, and this percentage decreases as the level of the application's technological complexity increases. Connectivity, mobile apps, digital platforms, software, global satellite positioning systems, remote sensing, and field sensors are the main technologies used.

Brazilian farmers' main perceptions in relation to benefits are linked to increased productivity, product higher selling potential, better planning, and management of production systems, 95% of farmers would like to learn more about new technologies and strengthen its applications. The use of such digital technologies has the potential to increase the sustainable management of natural resources (soil and water) and to reduce the use of agricultural inputs, making agricultural areas more productive and reducing their environmental impact. However, important difficulties to amplify its use were pointed out. The cost of purchasing machines, equipment and applications, problems with/or lack of connectivity in rural areas were the main issues presented. It is suggested that future studies focus on the new research questions from this work, more specifically on farmers' social and economic behavior towards adoption of new technologies and whether the adopters have a competitive advantage compared to non-adopters. Further research could also consider the way technology adoption will occur by the new generation, accordingly to the level of Internet access in the rural areas and hoping that they will continue to run the family business.

**Author Contributions:** Conceptualization, É.L.B., L.A.d.C.J., I.D.S. and A.L.J.; Methodology, É.L.B., L.A.d.C.J., I.D.S. A.L.J., C.C.d.C. and R.Y.I.; Software, C.C.d.C., D.d.C.V. and É.L.B.; Validation, É.L.B and A.L.J.; Formal Analysis, É.L.B., L.A.d.C.J., I.D.S., A.L.J., C.C.d.C., D.d.C.V., R.Y.I., C.R.G., V.R.F. and A.R.R.; Investigation, É.L.B., L.A.d.C.J., I.D.S. and A.L.J.; Resources, É.L.B., V.R.F. and A.R.R.; Data Curation, C.C.d.C. and D.d.C.V.; Writing—Original Draft Preparation, É.L.B., L.A.d.C.J., I.D.S A.L.J. and D.d.C.V.; Writing—Review & Editing, É.L.B., C.C.d.C., D.d.C.V., R.Y.I., C.R.G., V.R.F. and A.R.R.; Visualization, É.L.B.; Supervision; Project Administration and Funding Acquisition, É.L.B. and V.R.F. All authors have read and agreed to the published version of the manuscript.

**Funding:** This research article is part of a project that has received funding from the Brazilian Micro and Small Business Support Service (SEBRAE) and Brazilian Agricultural Research Corporation-Secretariat of Innovation and Business (EMBRAPA) under grant agreement N° 10200.18/0083-3. The opinions expressed in the article reflect only the authors' view.

**Conflicts of Interest:** The authors declare no conflict of interest.

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
