# Peer review of "Precision and Digital Agriculture: Adoption of Technologies and Perception of Brazilian Farmers"

_agriculture, doi:10.3390/agriculture10120653_

Round 1

Reviewer 1 Report

This article presents the results from a survey of Brazilian farmers regarding their use and attitudes on digitalisation and precision ag. The sample size is good and the research correctly conducted. While it may not provide very novel results, it may provide further understanding for the context of Brazil. 

I have some comments that could be improved or clarified:

  • Figure 2 where piecharts are used for showing percentages are not well designed. A percentage pie chart should add up to 100, but in this case there are overlaps so it does not, which makes the chart difficult to interpret. It is maybe better to use another type of visualisation.
  • I am unsure about having "internet and connectivity in the farm" as part of digitalisation/precision ag. This is more of a precondition than a technique/usage, specially as using internet for "information" and "management" are separate categories. Is there overlap of answers that have internet but do not use it for anything?. I would have presented this separately. 
  • In figure 6 there are several ones connected to cost, which is a bit confusing and there is no way of knowing if they answered several or all of the cost or just some of them.
  • It would have been interesting to compare the results of the bigger farms, >50ha to the smaller farms, as investment and usefulness is quite different in those situations.
  • Conclusions could be a bit more extensive in the discussion.

Author Response

Response to Reviewer 1 Comments

The authors are grateful for the suggestions / indications for improving the manuscript. The text was revised accordingly, with the adjustments indicated below and in the manuscript.

---

Point 1: Figure 2 where piecharts are used for showing percentages are not well designed. A percentage pie chart should add up to 100, but in this case there are overlaps so it does not, which makes the chart difficult to interpret. It is maybe better to use another type of visualisation.

Response 1: Adjusted p.6/181

Figure 2 was changed with another type of graph / visualization for cases with data above 100%.

Point 2: I am unsure about having "internet and connectivity in the farm" as part of digitalisation/precision ag. This is more of a precondition than a technique/usage, specially as using internet for "information" and "management" are separate categories. Is there overlap of answers that have internet but do not use it for anything? I would have presented this separately.

Response 2: The authors agree with the reviewer’s observation, i.e. internet and connectivity are pre-conditions to the use of other digital technologies. However, other works [14, 37] have considered connectivity/wireless as a form of technology. Thus, in this study and considering the brazilian characteristics, we decided to treat internet/connectivity as being a low-complexity technology (Table 1).

[14] McKinion, J.M.; Turner, S.B.; Willers, J.L.; Read, J.J.; Jenkins, J.N.; McDade, J. Wireless technology and satellite internet access for high-speed whole farm connectivity in precision agriculture. Agricultural Systems 2004, 81, 201-212. https://doi.org/10.1016/j.agsy.2003.11.002

[37] Zervopoulos, A.; Tsipis, A.; Alvanou, A.G.; Bezas, K.; Papamichail, A.; Vergis, S.; Stylidou, A.; Tsoumanis, G.; Komianos, V.; Koufoudakis, G.; Oikonomou, K. Wireless Sensor Network Synchronization for Precision Agriculture Applications. Agriculture 202010, 89. https://doi.org/10.3390/agriculture10030089

Adjusted p.3/131

Internet and Connectivity/Wireless...

Adjusted p.6/194

...involving internet access and connectivity/wireless...

Point 3: In figure 6 there are several ones connected to cost, which is a bit confusing and there is no way of knowing if they answered several or all of the cost or just some of them.

Response 3: Well observed. Each item on figure 6 was an option in the original questionnaire, where respondents could indicate one or more difficulties/challenges. Indeed, several of the items have a relation to cost such as training, acquisition, upgrades, services etc. However, in this question we choose to separate the different costs in order to evaluate if anyone had a larger response than the rest.

Adjusted p.11/351

Many of these challenges are related to the cost of digital technologies, be it purchase, upgrade or acquiring services.

Point 4: It would have been interesting to compare the results of the bigger farms, >50ha to the smaller farms, as investment and usefulness is quite different in those situations.

Response 4:

Adjusted p.12/364

When analyzing the questions related to farmers investment considering only smaller farms (> 20ha), it was observed that 68.5% indicated the value for the acquisition of machines, equipment, or applications; and 45.3% indicated the value for hiring specialized service providers as the main difficulties and challenges. Thus, acquisition costs are considered a large difficulty/challenge across all farm sizes. However, it is interesting to note that for properties above 500 ha, despite large percentage indicating acquisition costs as a challenge (66%), connection issues were the main answer (69%). This difference is even more pronounced on larger properties (> 1000 ha), where connection issues were indicated by 69% of the respondents and acquisition costs by 54%, still a large percentage.

Point 5: Conclusions could be a bit more extensive in the discussion.

Response 5:

Adjusted p. 13/417

In addition to the technologies already consolidated in precision agriculture in Brazil, such as field sensors, remote sensors, embedded electronics, telemetry and automation, digital agriculture also involves the use of mobile apps, digital platforms, social networks, big data, internet of things, artificial intelligence, cloud computing, blockchain and cryptography, allowing to support greater sustainability of production systems. The use of digital technologies, underscored by brazilian farmers, are differentiated and currently cover a wide spectrum, mainly in: obtaining information about property management, product sales, inputs acquisition, property mapping and climate forecasting, presenting future interests in a greater use in production estimation, detection and control in factors related to soil physic-biotics factors and other stresses of the agricultural crops.

It is important that the public and private sectors evaluate their forms of aid to enhance the use of these digital technologies, since a large proportion of farmers, especially those with lower technical skills, still need support and technical monitoring to adapt and become more familiar with the use and applications of these technologies. Moreover, it is essential to structure a support process to enable the producer´s decision making by presenting the costs and benefits in the acquisition and use of such technologies.

Adjusted p. 13/ 434

The digital agriculture can be understood as a set of technologies for communication, information and analysis that allows farmers to plan, monitor and manage the operational and strategic activities of the agriculture production systems, from pre-production, in production, and post-production. The results observed in the survey indicate that 84% of the Brazilian farmers interviewed use at least one digital technology in their production system, and this percentage decreases as the level of the application´s technological complexity increases. Connectivity, mobile apps, digital platforms, software, global satellite positioning systems, remote sensing and field sensors are the main technologies used.

Brazilian farmers´ main perceptions in relation to benefits are linked to increased productivity, product higher selling potential, better planning, and management of production systems, 95% of farmers would like to learn more about new technologies and strengthen its applications. The use of such digital technologies has the potential to increase the sustainable management of natural resources (soil and water) and to reduce the use of agricultural inputs, making agricultural areas more productive and reducing their environmental impact. However, important difficulties to amplify its use were pointed out. The cost of purchasing machines, equipment and applications, problems with/or lack of connectivity in rural areas were the main issues presented.

It is suggested that future studies focus on the new research questions from this work and more specific on farmers’ behavior, economic measure a single technology and compare the adopters have a competitive advantage compared to non-adopters. Further research could also consider the way technology adoption will occur by the new generation, accordingly to the level of internet access in the rural areas and hoping that they will continue to run the family business.

Reviewer 2 Report

Many publications have discussed the development of the innovative industry. The area concerning Industry 4.0 covers various sectors of the economy. One of them is agriculture. However, it is insufficiently considered from the standpoint of the Fourth Industrial Revolution, with digitization offering the opportunities for the rapid development not only of agriculture in Brazil but also of other countries in the world. The authors of the paper "Precision and Digital Agriculture: Adoption of Technologies and Perception of Brazilian Farmers", indicated the aspects of the use of the most advanced information technology by farmers in Brazil to close the still existing research gap in this area.

The authors discussed very interesting research, which may represent the basis for further investigations and comparisons on the development of digital agriculture in other countries.

The obtained research results are transparently presented to the reader and do not raise any objections to the achievement of the research goal set by the authors. They are illustrated both numerically in tables and in diagrams using appropriate statistical methods. However, I would suggest to the authors:

· In the Abstract section, it is necessary to specify in detail the research methods used and to provide information on the period when the research was conducted.

· Please indicate the percentage of 504 farmers surveyed in the total number of farmers in Brazil.

· In the Conclusions section, please attempt to interpret the results in the context of sustainable agricultural development in Brazil.

I hope that my suggestions for changes will contribute to making the publication even more readable to scientists and to those who see digitization as an opportunity for farmers' development.

Author Response

Response to Reviewer 2 Comments

The authors are grateful for the suggestions / indications for improving the manuscript. The text was revised accordingly, with the adjustments indicated below and in the manuscript.

---

Point 1: In the Abstract section, it is necessary to specify in detail the research methods used and to provide information on the period when the research was conducted.

Response 1:

Adjusted p.1/26

...

This article presents the results of a survey carried out with 504 Brazilian farmers about the digital technologies in use, as well as current and future applications, perceived benefits, and challenges. The questionnaire was prepared, organized and made available to the public through the online platform LimeSurvey and was available from April 17 to June 2, 2020. The primary data obtained for each question previously defined were consolidated and analyzed statistically.

...

Point 2: Please indicate the percentage of 504 farmers surveyed in the total number of farmers in Brazil.

Response 2:

Adjusted p. 4/151

Considering the size of the sample population of rural properties in Brazil approximately 1.5 million farmers in the survey profile, [38], it was stipulated to reach at least 385 questionnaires answered in full to obtain up to 5% margin of error with a 95% reliability level. However, results were gathered for 504 farmers (0.03% of population), a higher response rate than originally expected.

Point 3: In the Conclusions section, please attempt to interpret the results in the context of sustainable agricultural development in Brazil.

Response 3:

Adjusted p. 13/ 434

Brazilian farmers´ main perceptions in relation to benefits are linked to increased productivity, product higher selling potential, better planning, and management of production systems, 95% of farmers would like to learn more about new technologies and strengthen its applications. The use of such digital technologies has the potential to increase the sustainable management of natural resources (soil and water) and to reduce the use of agricultural inputs, making agricultural areas more productive and reducing their environmental impact.
